# Connexins in Cancer, the Possible Role of Connexin46 as a Cancer Stem Cell-Determining Protein

**DOI:** 10.3390/biom13101460

**Published:** 2023-09-27

**Authors:** Isidora M. León-Fuentes, María G. Salgado-Gil, María S. Novoa, Mauricio A. Retamal

**Affiliations:** Programa de Comunicación Celular en Cáncer, Facultad de Medicina Clínica Alemana, Universidad del Desarrollo, República de Honduras 12740, Las Condes, Santiago 7610496, Chile; ileonf@udd.cl (I.M.L.-F.); marisalgadog@udd.cl (M.G.S.-G.); mnovoab@udd.cl (M.S.N.)

**Keywords:** Connexin46, GJA3, cancer stem cells, breast cancer, gap junction channels

## Abstract

Cancer is a widespread and incurable disease caused by genetic mutations, leading to uncontrolled cell proliferation and metastasis. Connexins (Cx) are transmembrane proteins that facilitate intercellular communication via hemichannels and gap junction channels. Among them, Cx46 is found mostly in the eye lens. However, in pathological conditions, Cx46 has been observed in various types of cancers, such as glioblastoma, melanoma, and breast cancer. It has been demonstrated that elevated Cx46 levels in breast cancer contribute to cellular resistance to hypoxia, and it is an enhancer of cancer aggressiveness supporting a pro-tumoral role. Accordingly, Cx46 is associated with an increase in cancer stem cell phenotype. These cells display radio- and chemoresistance, high proliferative abilities, self-renewal, and differentiation capacities. This review aims to consolidate the knowledge of the relationship between Cx46, its role in forming hemichannels and gap junctions, and its connection with cancer and cancer stem cells.

## 1. Introduction

Cancer is a devastating non-communicable disease that claims millions of lives worldwide each year [1]. Despite significant clinical and scientific efforts, a cure for cancer remains elusive. One key factor contributing to this challenge is the development of resistance by some cancer cells to conventional treatments such as chemotherapy, radiotherapy, and immunotherapy [2,3,4,5]. Normally, cells in our body divide to repair and maintain healthy tissues, but in cancer, this process becomes disrupted, leading to uncontrolled cell division and the formation of solid tumors in most cases. As these tumors grow, certain cancer cells detach and spread to distant parts of the body through the bloodstream or lymphatic system, a complex process known as metastasis [6]. Although there are many types of cancer, they all share common features such as cell dedifferentiation, significant changes in cell metabolism, loss of contact with neighboring cells, and the presence of cancer stem cells (CSCs) [7,8]. Although CSCs make up a small fraction of cells within a tumor, they are critical to cancer progression, recurrence, metastasis, and resistance to treatment [9,10]. Similar to other types of stem cells, CSCs possess the ability to self-renew and differentiate, which are key characteristics known as stemness in cancer cells [11]. Additionally, CSCs exhibit a high level of drug resistance, making them particularly challenging to eliminate [12]. Hence, a proposed model for chemoresistance and cancer relapse posits that, while conventional chemotherapy induces the death of cancer cells, cancer stem cells (CSCs) manage to survive. Over time, these CSCs repopulate the tumor with both more CSCs and new cancer cells. Consequently, this leads to the development of resistance to chemotherapy within the entire tumor [13]. On the other hand, Connexins (Cxs) are transmembrane proteins involved in the progression of cancer. Some of these proteins inhibit cancer cell aggressiveness, while others exacerbate it. Despite extensive research, the precise role of these proteins in cancer biology remains incompletely understood. However, in the past decade, certain studies have begun to shed light on the potential role of a specific type of Cx, namely Cx46, as an enhancer of CSC characteristics. In this review, our primary objective is to explore the relationship between the presence of Cx46 and the aggressiveness of cancer, with a particular focus on how it contributes to the gain of function in CSCs. We also propose potential molecular mechanisms that may underlie this phenomenon. Additionally, we address some of the controversies that have arisen between basic research findings and clinical observations.

## 2. Connexins General Characteristics

Connexins (Cxs) are a family of proteins that share a common plasma membrane structure, characterized by four transmembrane domains, two extracellular loops, one intracellular loop, and both the C- and N-termini located on the cytoplasmic side [14] (Figure 1). In humans, 21 isoforms of Cxs have been identified [15], and these isoforms are named based on their predicted molecular weight (e.g., Cx46 is predicted to have a molecular weight of ~46 kDa). While Cx isoforms display significant homology, their C-terminus is the most variable region in terms of length and amino acid sequence. Accordingly, the C-terminus of each Cx type contains diverse regulatory sites, such as consensus phosphorylation sites [16,17], pH [18], protein–protein interaction sites [19,20], and cleavage sites [21,22], among others. With the exception of Cx23 [23], almost all Cxs have six conserved extracellular cysteines, which are thought to form intramolecular disulfide bridges, crucial for hemichannel docking and the formation of gap junction channels (GJCs) [24]. However, the role of these extracellular cysteines in hemichannel function seems to be more complex, thus Cx43 hemichannels formed by a protein without extracellular cysteines remain functional [25]; however, substitution of a single cysteine for alanine in Cx46 results in hemichannels with highly reduced permeability to DAPI [26]. Moreover, these extracellular cysteines have been proposed as redox sensors in the Cx46 hemichannel [27]. In mammals, almost all cell types express at least one type of Cx, although the level of expression varies among different Cx types. Among these, Cx43 is the most widely expressed [28,29,30,31]. On the other hand, Cx46 is predominantly limited to the eye lens [32,33]. Interestingly, despite the differences in regulation and expression observed among Cxs, all of them exert their physiological and pathological functions via three distinct mechanisms: hemichannel, GJCs, and engagement in protein–protein interactions. In the next sections, we will discuss each of these mechanisms of action with some degree of detail.

Hemichannels play a crucial role in cell-to-cell communication and are composed of six Cx monomers. It depends on the type of Cx, whether it forms in the endoplasmic reticulum, Golgi apparatus, or post-Golgi vesicles, and is subsequently transported to the plasma membrane [34,35]. The presence of undocked hemichannels on the plasma membrane has been demonstrated using various techniques such as Cryo-EM, biochemical assays, electrophysiological measurements, freeze-fracture, and functional assays including dye uptake and ATP release [36,37,38,39,40,41,42,43,44,45,46,47,48,49,50]. These hemichannels act as channels that are permeable not only to ions but also to molecules such as ATP [47], glutamate [48], glucose [41], and D-serine [50], among others (Figure 1). This property is based on the fact that hemichannels possess relatively large por456+es. For instance, the pore of Cx26 has an approximate diameter of 14 Å [51]. However, their selectivity appears to be complex and depends on molecular characteristics such as size, charge, and shape [52]. Currently, the understanding of the physiological roles of hemichannels is expanding each year. Thus, they have been implicated in synaptic regulation [53], memory consolidation [54], and the release of neuroactive molecules [42,55]. Furthermore, they have been found to be involved in osmotic regulation [56], light processing in the retina [57], CO_2_ sensing [58], and PGE_2_ release [59]. Many of these functions are enabled by the release of signaling molecules, as previously discussed. However, it is important to note that the controlled opening of hemichannels is crucial for these physiological processes, as uncontrolled opening can lead to cell damage due to excessive entry of Na^+^ and Ca^2+^ [60,61,62] or even cell lysis [63]. Despite the significance of controlled hemichannel opening, the precise mechanisms that regulate them under such conditions remain not yet fully understood. One possible regulatory mechanism involves transient elevations of intracellular Ca^2+^ concentration [64,65]. Further research is needed to fully understand the hemichannel control mechanisms in the physiological contexts.

GJCs form when two hemichannels dock at the junctional membrane between adjacent cells, where each hemichannel is contributed by different cells. As hemichannels, GJCs enable the passive exchange of ions and small molecules, facilitating communication between the cytoplasm of neighboring cells [14]. Several molecules have demonstrated some degree of permeability via GJCs, including second messengers (i.e., cAMP and IP3), metabolites (i.e., glucose), and even small peptides [66,67,68] (Figure 1). However, the solute selectivity of GJCs is determined by specific Cx isoforms [69,70]. For instance, Cx32 GJCs have been shown to be approximately 100 times less permeable to ATP than Cx43 GJCs, but they exhibit more effective transferring of adenosine compared to Cx43 GJCs [71]. As hemichannels, the variation in GJC permeability between Cx isoforms is likely attributed to the size, shape, and charge of the ions or molecules passing through these channels [72]. Under physiological conditions, GJCs play a crucial role in coordinating metabolic and signaling responses among groups of cells. For instance, it is widely accepted that Cx43 GJCs coordinate the flow of action potentials between cardiomyocytes, ensuring proper heart rhythm and function [73]. However, when changes in the permeability of Cx43 GJCs may occur due to a health problem such as a heart attack, the conduction of action potentials across the GJCs is hindered, leading to slowed conduction and the potential emergence of arrhythmias [74,75]. It is important to note that Cx43-mediated arrhythmias are a complex phenomenon that extends beyond alterations in ion conduction via GJCs. It has been also demonstrated that Cx43 can form hemichannels that can open to the plasma membrane, allowing ion flux into the extracellular space and affecting cardiomyocyte excitability [76]. Consequently, Cx43 hemichannels have been proposed as potential targets for the arrhythmia treatment [74,77]. Another example is Cx46 and Cx50 GJC in the eye lens, where they play a crucial role in the metabolic maintenance of lens cells [78]. The lens, a transparent structure without blood supply, relies on an intercellular circuit facilitated by Cx46 and Cx50 GJCs to ensure the exchange of nutrients, oxygen, and metabolic waste [79,80,81]. Specifically, Cx46 GJCs are instrumental in enabling the transport of reduced glutathione (GSH) from cortical fiber cells to nuclear cells via diffusion [82]. Consequently, any mutations or oxidative stress affecting Cx46 and Cx50 can lead to alterations in GJCs’ properties and may contribute to cataract formation [83,84]. These two cases exemplify the role of GJCs in physiological conditions, which is the exchange of metabolites and signaling molecules for the correct cell functioning and ultimately the tissue, as well as the organ. Alterations in this communication are usually associated with diseases or, in extreme cases, cell death.

In addition to the canonical functions of Cxs in forming hemichannels and GJCs, it is crucial to acknowledge that Cxs can also exert cellular effects in a channel-independent manner. Cxs have the ability to physically interact with several proteins. Based on the available data, nowadays it is possible to suggest two types of protein–protein interaction mechanisms involving Cxs. The first mechanism involves interactions that encompass the entire Cx protein, allowing it to mainly interact with proteins located near or within the plasma membrane. The second mechanism involves protein–protein interactions between the Cx-free C-terminal and various cytoplasmic proteins. This type of interaction presents an exciting topic of research and opens up new possibilities for understanding additional cellular functions mediated by Cxs.

As illustrative examples of the first proposed mechanism, Cx43 has been found to interact with β-catenin at the plasma membrane of cardiomyocytes [85]. This interaction involves the Cx43 C-terminal and is inhibited by a Src-mediated Cx43 phosphorylation at Y265 and Y313 [86]. Interestingly, this interaction plays an important role in sequestering β-catenin at the plasma membrane, resulting in a reduction in Wnt/β-catenin signaling pathway strength [85]. Another significant instance involves the interaction between Cx43 and ZO-1, which is mediated by the Cx43 C-terminal and the second PDZ domain of ZO-1 [87,88]. Notably, this protein–protein interaction appears to be regulated during the cell cycle in a rat kidney cell line (NRK) [88] and plays a pivotal role in controlling Cx43 GJC formation [89]. Furthermore, building upon this knowledge, a mimetic peptide called aCT-1 has been developed, which mimics the specific segment responsible for the interaction between Cx43 and ZO-1. This peptide has shown promising applications in skin wound healing [31,90], where its potential to modulate cellular interactions and signaling pathways holds great therapeutic value. On the other hand, Cxs have been detected in the mitochondria of various cell types, such as cardiomyocytes [91], mice vascular endothelial cells [92], and rat retinal endothelial cells [93]. Notably, mass spectrometry analyses of mice cardiomyocyte mitochondria have uncovered intriguing interactions between Cx43 and apoptosis-inducing factor (AIF) as well as the β-subunit of the electron transfer protein (ETFB) [94]. These findings strongly suggest that Cx43-mediated protein–protein interactions are not confined to specific cell membranes but can be observed in different cellular contexts. The presence of Cxs within mitochondria adds another layer of complexity to their functional roles beyond their well-established involvement in cell-to-cell communication. Understanding the significance of these protein interactions in the mitochondrial compartment could offer novel insights into mitochondrial function, cellular signaling, and cell fate regulation.

Regarding the second proposed mechanism, compelling evidence has demonstrated that the Cx C-terminal region can be transcribed independently from the rest of the Cxs [95,96], facilitated by an mRNA internal ribosome entry site (IRES) [97]. This unique C-terminal peptide demonstrates the ability to establish its own protein interactions [98,99]. In this context, the free C-terminal region of Cx43 has been shown to increase the migratory capacity of glioma cells via its interaction with the actin cytoskeleton [100]. Conversely, expression of the Cx43 C-terminal in U2OS (an osteosarcoma cell line) and HeLa cells were found to decrease their rate of cell division [101,102], potentially via its interaction with S-phase kinase-associated protein 2 (Skp2) [101]. Similarly, a peptide derived from the Cx43 C-terminal (TAT-Cx43 266-283) has been demonstrated to reduce the cancer stem cell (CSC) phenotype by inhibiting c-Src in patient-derived glioma models [103,104] and in vivo mouse models [104]. Moreover, a peptide encompassing the Cx43 C-terminal (285–363) interacts with Akt within its pleckstrin homology (PH) domain, leading to the inhibition of Akt’s function [105]. Additionally, the Cx C-terminal also plays a role in regulating the expression of various proteins [106], including N-cadherin [107] and p53 [108], as well as certain miRNAs [108]. The suggested mechanism by which the Cx C-terminal region exerts its transcriptional regulation is via its localization within the cell nucleus [109,110], where it can modulate the activity of nuclear proteins [111,112]. In the case of Cx43 C-terminal, it is proposed to act directly as a transcription factor [107].

These findings highlight the multifaceted nature of the Cx C-terminal region, extending beyond its traditional role in Cx-mediated communication. Understanding the diverse protein interactions and transcriptional regulatory functions of the Cx C-terminal region provides valuable insights into the broader cellular mechanisms orchestrated by Cxs. Further investigations in this area hold significant potential for uncovering novel therapeutic targets and strategies for various diseases and conditions associated with Connexin dysfunction. For more details on Cx-based protein–protein interaction, see previous reviews [20,98,112].

## 3. Cx46 in Cancer

In the 1950s, the idea that Connexins (Cxs) might influence the cell division rate was introduced [113]. Early observations indicated that the downregulation of Cx32 and Cx43, along with the loss of GJC-mediated communication, correlated with neoplastic activity in the liver and brain [114,115,116]. However, as research has progressed, it has become evident that the role of Cxs in cancer is far more complex, with some Cxs exhibiting pro-tumorigenic effects while others demonstrate anti-tumorigenic properties [117,118]. Despite numerous studies that have associated the expression of specific Cxs with changes in cancer cell characteristics, such as cell division rate, cell migration, and the CSC phenotype [118,119], only a limited number of these investigations have successfully identified the molecular mechanisms underlying the involvement of Cxs in cancer. Understanding the precise molecular mechanisms through which Cxs impact tumorigenesis may offer crucial insights into the development of targeted therapeutic approaches and potential biomarkers for various types of cancer.

Similar to many other Cxs, Cx46 has the ability to form both functional GJCs and hemichannels. In the case of Cx46-mediated GJCs, they exhibit a conductance ranging from 148 to 192 pS when studied in N2A cells, and they display sub-conductances in the range of 10 to 60 pS [120,121]. In contrast, when considering hemichannels formed by Cx46, recordings indicate a conductance of approximately 250–300 pS [122,123,124] with a sub-conductance of about 40 pS [122]. Regarding their ionic permeabilities, Cx46 hemichannels show a preference for cations over anions [124]. In terms of the permeability to larger molecules, Cx46 expressed in Xenopus oocytes was shown to allow the passage of carboxyfluorescein, Lucifer Yellow, and Ethidium [125], whereas Cx46 expressed in HeLa cells has demonstrated permeability to DAPI [126]. Additionally, Cx46-based GJCs appear to facilitate the flux of the antioxidant molecule GSH (glutathione) between lens cells, indicating a role in the transport of this vital cellular component [82]. In summary, it is evident that Cx46-based channels share several characteristics with channels formed by other connexins, highlighting the versatility and commonalities in Cx channel behavior across various contexts.

For many years, the attention on Cx46 was only focused on its role in physiological and pathological phenomena in the eye lens [33,79,84]. However, since the 2000s, the study of its possible role in cancer progression began. These studies involved animal models of lung and bone cancer, which revealed a significant decrease in both Cx46 mRNA and protein levels [127,128]. These findings strongly suggested that Cx46 may have a potential anti-cancer role. However, in 2010, a groundbreaking article was published, associating Cx46 with human cancer [129]. This study demonstrated a significant increase in Cx46 levels, as measured by Western blot and immunohistochemistry in samples of human infiltrating breast carcinoma [129]. Furthermore, in a mouse xenograft model, the injection of siRNA against Cx46 inhibited the growth of the MCF-7 human breast cancer cell line [129] and the Y79 human retinoblastoma cell line [130]. Interestingly, Cx46 is expressed in the lens, a hypoxic tissue characterized by its limited blood irrigation. This shared hypoxic condition between the lens and certain solid tumors, such as breast cancer, led to the initial hypothesis that Cx46 might function as a protective factor against hypoxia. To test this hypothesis, Banerjee et al. overexpressed Cx43GFP and Cx46GFP in N2A cells (a mouse neuroblastoma cell line that does not express any type of Cx) and subjected these cells to 1% oxygen (hypoxia). After 24 h in this condition, approximately 90% of the N2A wild-type and N2A cells transfected with Cx43GFP died, while the percentage was only 50% in cells transfected with Cx46GFP [129].

As previously mentioned, Cx46 has been implicated in promoting tumor growth in xenograft models [129,130]. Interestingly, only CSCs have the unique ability to initiate tumor growth in xenograft models [131,132], which suggests a potential link between Cx46 and the CSC phenotype. Supporting this idea, studies have exclusively identified Cx46 expression in CSCs of human glioblastoma, where it plays a crucial role in self-renewal [133]. Given the limited understanding of the underlying mechanism, our laboratory has been investigating whether Cx46 enhances the CSC phenotype in other types of cancer cells. Our research findings indicate that MCF-7 cells expressing Cx46GFP display elevated levels of Sox2 and Oct4 mRNA, along with the formation of larger tumorspheres and clonogenic colonies when compared to Cx46-negative MCF-7 cells [134]. These observations strongly suggest that Cx46 acts as a pro-tumorigenic factor, although the precise mechanism remains elusive. However, Acuña et al. recently shed some light on this matter by demonstrating that MCF-7 cells expressing Cx46 release more exosomes containing Cx46 than Cx46-negative MCF-7 cells [135]. Of greater significance, the Cx46 present on the exosomal membrane enhances the transfer of critical “information” between the exosome and the recipient cell. These findings suggest that Cx46-mediated exosomal communication may play a role in promoting the CSC phenotype and tumorigenic behavior. Additional research is needed to fully understand the molecular mechanisms by which Cx46 affects CSC function and contributes to tumor progression.

## 4. Unraveling the Mechanisms of Action of Cx46 in Cancer Cells

### 4.1. The Possible Role for Cx46-GJCs

In general, it is well accepted that Cxs do not form functional GJCs in cancer cells [136] and that their re-expression and formation of GJCs can reduce cancer cell proliferation and tumor formation in vivo [137,138,139]. The most plausible mechanism for this anti-tumoral effect is allowing the diffusion of second messengers through them [140,141,142]. For example, the cAMP flow through Cx26- and Cx43-GJCs is associated with a decrease in cell division [143,144]. Similarly, Cx43 expression and GJC formation in lung cancer cells inhibit the CSC phenotype [145]. However, GJCs are not always associated with anti-tumoral effects. Thus, Cx43-GJCs expressed in glioma cells increase their invasiveness via the exchange of miRNAs between glioma cells and astrocytes [146]. In the case of Cx46, its expression was strongly associated with CSC self-renewal, and propagation in human glioblastoma [133]. Interestingly, a molecule that inhibits Cx46-GJCs (clofazimine) reduced the CSC phenotype [147], suggesting an important role of Cx46-GJCs in the maintenance of CSCs (Figure 2).

### 4.2. The Possible Role for Cx46 Hemichannels

The role of hemichannels in cancer has been poorly investigated. As previously mentioned, Cx hemichannels enable the exchange of signaling molecules between the cytoplasm and the extracellular environment [38,148,149]. For example, Cx43 hemichannels regulate H9c2 and 3T3 cell proliferation via ATP [150] and NAD^+^ [151] release. Additionally, it has been proposed that Cx43 hemichannels can activate Akt in NRK-E52 cells via the release of ATP [152]. Similarly, only functional Cx37 hemichannels have been found to suppress cell proliferation in rat insulinoma [153]. Regarding Cx46 hemichannels, their permeability to biological molecules like ATP has been suggested [154] but has not yet been proven. Because Cx46 hemichannels are permeable to synthetic molecules with molecular weight and size comparable to those of the biological molecules listed above [126,155,156], there is no reason to expect that Cx46 hemichannels are not permeable to them. Therefore, Cx46 hemichannels could increase the CSC phenotype via the exchange of biological molecules between the cytoplasm and the extracellular milieu (Figure 3).

### 4.3. The Possible Role of Cx46 Protein–Protein Interactions

As previously mentioned, Cxs can have intracellular effects that are independent of their role as channels, mainly via interactions with other proteins [20,112,157,158,159,160,161,162,163,164,165,166,167], and their C-terminus mediates the vast majority of these Cx–protein interactions [19,157,158,162]. Regarding CSCs, the Cx26 C-terminal interacts with Nanog, promoting CSC renewal in triple-negative breast cancer cells [164]. Likewise, the accumulation of Cx32 in the cytoplasm has been shown to enhance CSC renewal in HuH7 hepatoma cells [163], likely via protein–protein interactions. On the other hand, Cx26, which has been associated with a pro-tumorigenic role [118] increases PI3K/Akt activity in NSCLC cells in a channel-independent way [164]. Conversely, Cx43, which, in general, is considered anti-tumorigenic [145,167] via the interaction of its C-terminal with Akt [106], induces its inhibition [167]. Despite that, Cx46 can interact with several proteins [168,169,170,171,172], but the potential interaction between Cx46 and cancer-relevant proteins such as PI3K/Akt has not been investigated.

## 5. Cx46 in Human Cancer Samples

Unfortunately, only a few studies have investigated the expression of Cx46 in human breast cancer and its association with patient survival. Remarkably, utilizing fluorescent microscopy revealed that Cx46 expression in human breast cancer samples is significantly associated with improved overall survival (OS) for patients [173]. Nevertheless, other studies have found no significant correlation between Cx46 mRNA levels and patient OS [174]. Such contradictions between cell culture studies, where the presence of Cx46 is linked to a more aggressive phenotype of breast cancer cells and the results obtained from human samples warrant further investigation. A possible explanation for these conflicting results is that there might not be a direct correlation between Cx46 mRNA and protein levels. This possibility is supported by the fact that at least in myeloid leukemia cells, there is no correlation between the mRNA and protein levels for Cx26, Cx32, Cx37, Cx43, and Cx45 [175]. Additionally, in HUVEC cells exposed to laminar flow, there is an increase in Cx40 mRNA levels with no significant changes in protein levels [176]. Additionally, from our own laboratory experience, we recognize the crucial importance of selecting an appropriate and validated antibody to ensure reliable and representative results when Western blot analyses and immunofluorescence studies are performed. Therefore, in-depth and comprehensive experiments are indispensable to gain a clear understanding of the significance of Cx46 in breast cancer. By conducting further research, including larger clinical studies and meticulous examination of Cx46 expression at both the mRNA and protein levels, we can hope to unravel the complex role of Cx46 in breast cancer more accurately.

## 6. Discussion

In the last years, the role of Cx46 in cancer has risen, mostly because results in animal models and human cell lines point out that this protein could be pro-tumorigenic and, moreover, it can be a key factor for the enhancement of EMT and CSC in cancer cells. However, studies correlating human samples of patients with breast cancer suggest that the presence of Cx46 is linked to better overall survival. Therefore, further studies are needed to determine whether Cx46 protein levels are associated with a favorable or unfavorable patient prognosis. Studies correlating the levels of Cx46 mRNA with protein levels could shed some light and obtain better antibodies to help to determine the real Cx46 protein levels in human samples.

## Figures and Tables

**Figure 1 biomolecules-13-01460-f001:**
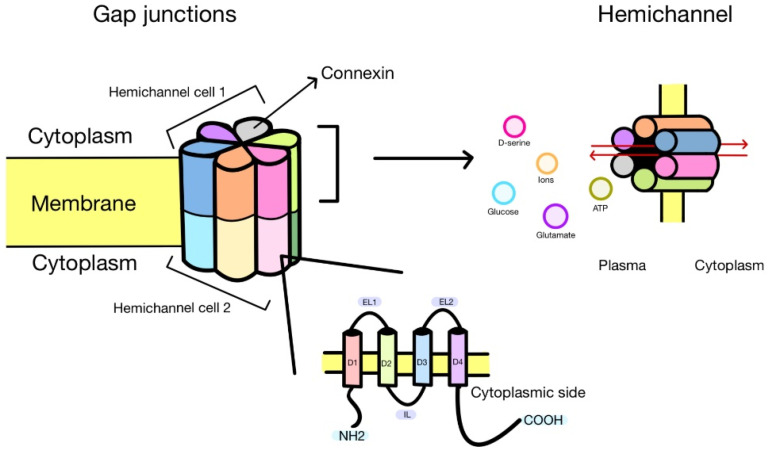
When two hemichannels from different cells come into contact, they form a gap junction channel, facilitating the flow of molecules and ions between these cells. Each hemichannel is composed of six connexins, each consisting of four transmembrane domains and three loops. One loop is intracellular (IL), while the other two are extracellular (EL1–EL2). D1–D4 denote transmembrane segments of a Cxs. Additionally, both the NH_2_ and COOH terminals face the cytoplasm. Furthermore, each hemichannel allows the bidirectional exchange (red arrows) of ions and molecules between the intracellular and extracellular environments.

**Figure 2 biomolecules-13-01460-f002:**
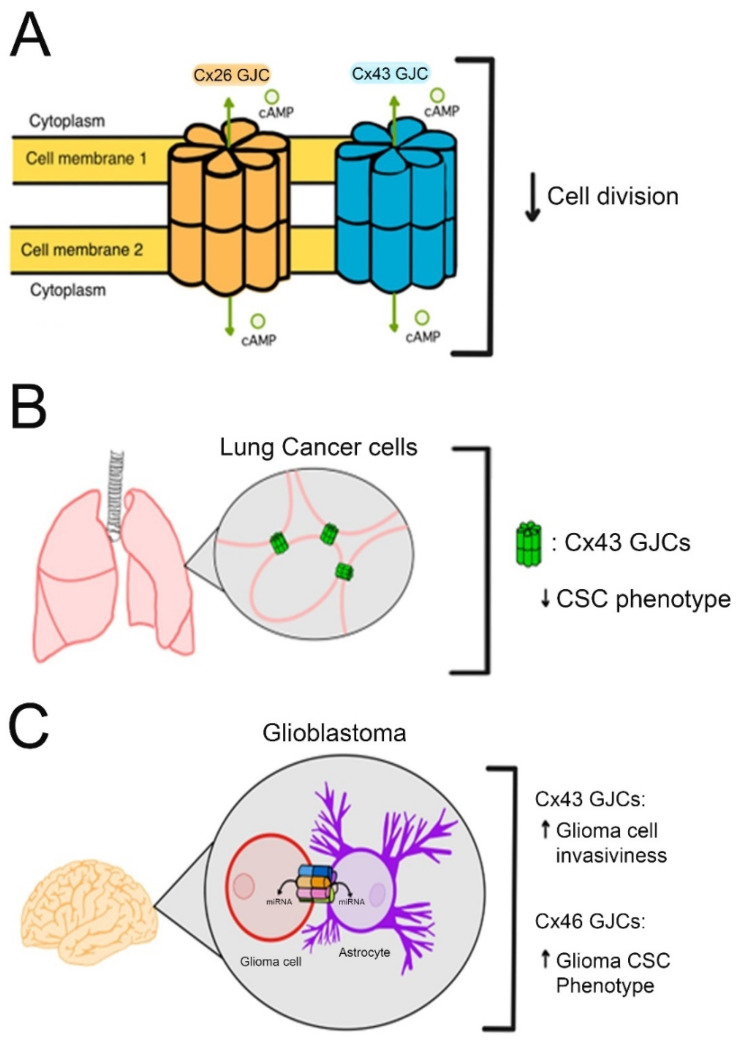
Role of Cx-GJCs in regulating cancer cell aggressiveness. (**A**) illustrates the relationship between the flow of cyclic adenosine monophosphate (cAMP) via connexin 26 (Cx26) and connexin 43 (Cx43) gap junction channels (GJCs) and its impact on cell division. (**B**). shows that the formation of Cx43 GJCs in lung cancer cells has an inhibitory effect on the Cancer Stem Cell (CSC) phenotype. (**C**) However, in glioblastoma, Cx43 GJCs are involved in the increase in glioma cells’ invasiveness capacity by facilitating the exchange of microRNAs (miRNAs) between glioma cells and astrocytes. Additionally, Cx46 GJC seems to enhance CSC phenotype, however, the mechanism remains unknown.

**Figure 3 biomolecules-13-01460-f003:**
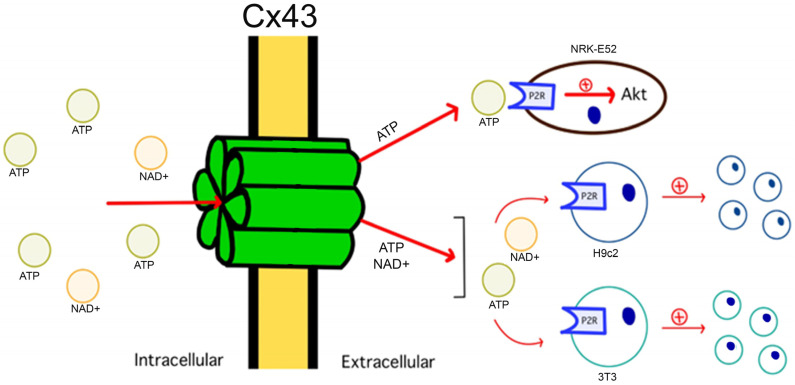
Role of Cx-Hemichannels in cell division regulation via the release of signaling molecule Release. Cx43 hemichannels are regulators of cell division by allowing the controlled release of signaling molecules, such as ATP and NAD^+^. These molecules, upon reaching the extracellular space, activate specific receptors and initiate signaling cascades, such as those controlled by Akt, which in turn modulates cell proliferation and impacts tissue homeostasis and development.

## Data Availability

We will send all the figures and results as requested.

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
