# Peer review of "Connexins in Cancer, the Possible Role of Connexin46 as a Cancer Stem Cell-Determining Protein"

_biomolecules, 2023, doi:10.3390/biom13101460_

Round 1

Reviewer 1 Report

In this review, the authors provide an update on the current knowledge regarding the role of Cx46 in cancer, with special focus on the association between the expression of this Cx and the aggressiveness of cancer, particularly in relation to the gain of function in cancer stem cells (CSCs). The Retamal group has been leading diverse studies in Cx46 in cancer. The text is well-written, its structure follows a logical, coherent and systematic sequence of the topics to be discussed and offers clear conceptualizations; for instance, it includes channel-dependent (as gap junction channels or hemi-channels) and channel–independent (ie., protein-protein interactions) mechanisms underlying the pro-tumorigenic properties of Cx46 and describe the knowledge gaps of Cx46 in patient prognosis and therapeutically strategies.

In addition, and importantly, due to the existing limited investigations concerning the role of Cx46 in cancer, authors present hypothesis and current contradictory findings to be resolved in future studies and they illustrate their proposals using examples of the widely expressed Cx43.

Concerns:

1 -  Introduction: it would be expected that authors contextualize the review before finishing the paragraph so that, those previous concepts naturally drive the reader to the aim of the revision: Cx46 and cancer. May be, they can describe more general aspects about Cxs in cancer before the last sentence “In this review, we aim to explore ……. in relation to the gain of function in CSCs”. As it is now, it seems that the topic will be transporters ABC and cancer.

2 -  Lines 79 – 80 old version and 78-79 new version:…… and functional assays including dye uptake [39–49]… include ATP release as another functional assay.

3 – Describe permeability and conductance properties of Cx46 GJs and Cx46 HCs if reported.

4 – Line 310 old version or 304 new version: Regarding the title “A possible of Cx46 protein-protein interactions “, it seems as if a word is missing.

The language is very correct.

Author Response

Concerns:

1 -  Introduction: it would be expected that authors contextualize the review before finishing the paragraph so that, those previous concepts naturally drive the reader to the aim of the revision: Cx46 and cancer. May be, they can describe more general aspects about Cxs in cancer before the last sentence “In this review, we aim to explore ……. in relation to the gain of function in CSCs”. As it is now, it seems that the topic will be transporters ABC and cancer.

R: Dear reviewer, we sincerely appreciate the time and effort you've dedicated to reviewing our work.  We think that you are completely right, and we modify our text, and now reads as follow: “Hence, a proposed model for chemoresistance and cancer relapse posits that while conventional chemotherapy induces the death of cancer cells, cancer stem cells (CSCs) manage to survive. Over time, these CSCs repopulate the tumor with both more CSCs and new cancer cells. Consequently, this leads to the development of resistance to chemotherapy within the entire tumor [13]. On the other hand, Connexins (Cxs) are transmembrane proteins involved in the progression of cancer. Some of these proteins inhibit cancer cell aggressiveness, while others exacerbate it. Despite extensive research, the precise role of these proteins in cancer biology remains incompletely understood. However, in the past decade, certain studies have begun to shed light on the potential role of a specific type of Cx, namely Cx46, as an enhancer of CSC characteristics. In this review, our primary objective is to explore the relationship between the presence of Cx46 and the aggressiveness of cancer, with a particular focus on how it contributes to the gain of function in CSCs. We also propose potential molecular mechanisms that may underlie this phenomenon. Additionally, we address some of the controversies that have arisen between basic research findings and clinical observations.

2 -  Lines 79 – 80 old version and 78-79 new version:…… and functional assays including dye uptake [39–49]… include ATP release as another functional assay.

R: Dear reviewer, thank you for your suggestion, it was incorporated.  

3 – Describe permeability and conductance properties of Cx46 GJs and Cx46 HCs if reported.

R: Dear reviewer, thank you for your suggestion, Cx46-based channels permeability and conductance were incorporated.  New text reads as follows “Similar to many other Cxs, Cx46 has the ability to form both functional GJCs and hemichannels. In the case of Cx46-mediated GJCs, they exhibit a conductance ranging from 148 to 192 pS when studied in N2A cells, and they display sub-conductances in the range of 10 to 60 pS [125,126]. In contrast, when considering hemichannels formed by Cx46, recordings indicate a conductance of approximately 250-300 pS [127–129], with a sub-conductance of about 40 pS [127]. Regarding their ionic permeabilities, Cx46 hemichannels show a preference for cations over anions [129]. In terms of the permeability to larger molecules, Cx46 expressed in Xenopus oocytes has been shown to allow the passage of carboxyfluorescein, Lucifer Yellow, and Ethidium [130], whereas Cx46 expressed in HeLa cells has demonstrated permeability to DAPI [131]. Additionally, Cx46-based GJCs appear to facilitate the flux of the antioxidant molecule GSH (glutathione) between lens cells, indicating a role in the transport of this vital cellular component [132]. In summary, it is evident that Cx46-based channels share several characteristics with channels formed by other connexins, highlighting the versatility and commonalities in Cx channel behavior across various contexts. “

4 – Line 310 old version or 304 new version: Regarding the title “A possible of Cx46 protein-protein interactions “, it seems as if a word is missing.

R: Thank you, problem was fixed.  

Reviewer 2 Report

In the manuscript,Isidora et al., reviewed the role of Cx46 in cancer. Cx46 is a protein that is involved in gap junction intercellular communication. While traditionally known for its function as an ion channel, recent studies have suggested that Cx46 may play a role in cancer, particularly in the context of Cancer Stem Cells (CSCs) and epithelial-mesenchymal transition (EMT). Studies conducted on animal models and human cell lines have indicated that Cx46 could have a pro-tumorigenic effect and contribute to the enhancement of EMT and CSCs in cancer cells. However, there have been contradictory findings in studies involving human samples, where the presence of Cx46 has been associated with improved overall survival in breast cancer patients.

The Prof. Retamal 's laboratory has been engaged in the research of Connexins protein for a long time and has done a good job. This review summarizes the functions of Cx46 in Cancer Stem Cell, and it is recommended to accept it.

Minor editing of English language required

Author Response

R: We extend our heartfelt gratitude for the time and effort you have invested in reviewing our work. Your kind words of encouragement and support are truly appreciated.

Reviewer 3 Report

The author provide a genral rreview on connexin and cancer. the authors are right to insinst on the lack of a clear mechanistical link between CX and th cancer,

my concern with the paper is thathe title is very restrictive and suggest a review  concentrated solely on Cx46

therefore i recommend to restrain the review on cx46 as suggested by the title

Author Response

R: Dear reviewer, we sincerely appreciate the time and effort you've dedicated to reviewing our work.  As both you and other reviewers have rightly pointed out, the existing knowledge base on Cx46 in the context of cancer is indeed restricted. To address this limitation and shed light on potential mechanisms, we have chosen to draw insights from other connexins where relevant. Therefore, we changed the title as follow “Connexins in cancer, possible role of Connexin46 as a Cancer Stem Cell-determining protein”. We believe that this new title more accurately reflects the essence of the manuscript.